# Evaluation of Ear Thermographic Imaging as a Potential Variable for Detecting Hypocalcemia in Postpartum Holstein Dairy Cows

**DOI:** 10.3390/ani15142055

**Published:** 2025-07-11

**Authors:** Guilherme Violin, Nanako Mochizuki, Simon Stephen Abraham Warju, Megumi Itoh, Takahiro Aoki

**Affiliations:** Department of Veterinary Medicine, Obihiro University of Agriculture and Veterinary Medicine, Obihiro 080-8555, Hokkaido, Japan; guilhermeviolin77@gmail.com (G.V.);

**Keywords:** hypocalcemia, infrared thermography, Holstein, dairy cattle, ear temperature, air temperature

## Abstract

Hypocalcemia is a metabolic disease common in dairy cattle during the first days after parturition. It impacts the welfare of those animals by causing apathy, making them unable to stand up, increasing the risk of other diseases, and if not treated quickly, may lead to death. One common symptom of hypocalcemia is the decrease in temperature of the ears, which is usually felt by hand, but this symptom still lacks research on how useful it can be for diagnosing this disease. This study found that surface ear temperature measured through infrared thermography has the potential of being a useful variable in identifying hypocalcemic cows, including subclinical cases. However, air temperature strongly influences the surface temperature of the ear, with colder ambient temperatures generating more accurate models than warmer ones.

## 1. Introduction

The modern dairy industry has greatly improved milk yield and quality through genetic selection and cow management. However, since these improvements primarily focus on milk yield—the most significant profit unit—dairy cows face numerous physiological challenges to sustain this increased production [1]. Among those challenges is hypocalcemia, also known as milk fever, which occurs due to the increased demand of calcium (Ca) for milk production. This leads to a decrease in available ionized Ca (iCa2+) in the blood stream that the cow is often unable to compensate through its homeostatic mechanisms [2].

Hypocalcemia is more common during the first 72 h postpartum and may be either clinical or subclinical. The clinical form is defined by the presence of characteristic symptoms such as paresis, lack of appetite, and cold extremities (feet and ears) [3,4], while the subclinical form is defined by a lack of visible symptoms, despite both being below the serum Ca threshold of a normocalcemic cow [5]. Subclinical hypocalcemia has also been reported to increase the risk of disorders such as retained placenta, displacement of abomasum and metritis [4].

The textbook literature outlines the physiological range for total serum Ca concentration as 2.42 to 3.09 mmol/L [6], but one study with clinically normal cows reported a mean concentration of total serum Ca of 2.25 (0.01) mmol/L (mean [SEM]) [7]. While both forms of hypocalcemia are usually characterized when blood serum Ca concentrations are below 2.00 mmol/L [4,5,8], different studies have used different thresholds of serum Ca concentration in order to define a cow as hypocalcemic. Caixeta et al. (2017) classified cows as hypocalcemic when their serum total Ca concentrations were ≤2.15 mmol/L [9], while Neves et al. (2017) reported different thresholds according to the days in milk (DIM) of the cow: ≤2.1 mmol/L for 3 DIM and ≤2.15 mmol/L for 2 and 4 DIM [10]. Gild et al. (2015) classified cows as hypocalcemic when their total Ca levels were below 1.87 mmol/L after correcting it to the albumin concentration of the animals [11]. However, serum concentrations of iCa2+, the bioactive form of Ca in the blood, has been reported to be a more accurate predictor for postpartum Ca status in Holstein dairy cows [12]. Cows with serum iCa2+ concentrations below 1.00 mmol/L were reported to be hypocalcemic [13,14,15]. The gold standard for diagnosing milk fever is still blood analysis, which when performed in a laboratory may take a considerable amount of time for the results, and can be costly when performed with a cow-side method such as the i-Stat [14].

One of the mechanisms of thermoregulation in mammals is conductive heat loss, which is controlled by the sympathetic regulation of blood vessels on cutaneous tissues. In response to cold, peripheral blood flow is diminished by vasoconstriction, limiting heat transfer between the organism and the environment. This vasoconstriction due to cold exposure happens first in the ears, followed by the lower extremities, and then through the skin of the trunk region [16]. The mechanism through which the ears of hypocalcemic cows become colder is still unclear, but it has been mentioned in the literature that when a cow is hypocalcemic, its heart musculature loses contractility, resulting in a diminished ability to thermoregulate, leading to the ambient exerting a bigger influence in skin temperature, particularly on the ears [17].

Although textbooks generally state that a decrease in skin temperature is a clinical symptom of milk fever, as far as the authors know, there is only one study that reported the use of ear skin temperature for detecting hypocalcemia [8]. In that study, an infrared thermometer was used to measure the temperature of the ear skin. The area analyzed was circular in shape and 2 cm in diameter, and was located between the tip and the middle part of the ear. Measurements were taken from both the front and the rear of the ear, and it was not completely clear if the temperature used in the analysis was the average of both measurements, or if they were compared to one another. In that study, air temperature was reported to be a confounding factor. In its conclusion, the study reported a sensitivity of 0.493 and a specificity of 0.738 when using ear skin temperature for detecting subclinical hypocalcemia. Based on these results, the authors did not recommend it as a reliable diagnosis method.

Since the publishing of that report, there have been new advancements in the field of infrared thermography. It is now possible to perform a more flexible and in-depth analysis of the photographed area, which could result in a better method than simple infrared thermometry, while also providing a visual insight into the temperature profile of the ear. Since the ears of hypocalcemic cows are more influenced by the environment ([6], p. 1259), the authors of the present study hypothesized that the difference between their ear surface temperature and that of normocalcemic cows would be greater in colder environments than in hotter ones, with the difference between air temperature and ear surface temperature being smaller in hypocalcemic cows than in normocalcemic ones. Consequently, data on ear surface temperature could prove to be more useful and accurate for the detection of hypocalcemia in colder air temperatures.

In order to build upon this past study, adapting the sampling method to infrared thermography and factoring air temperature in the analysis could prove to be fruitful changes. Thus, the present study aims to assess the viability of using infrared thermography of the ear for detecting hypocalcemia, both clinical and subclinical, in Holstein dairy cows divided into two separate air temperature groups: the lower temperature group (LT), with air temperatures ranging from −1.6 to 14.6 °C; and the higher temperature group (HT), including temperatures from 15.3 to 31.2 °C.

## 2. Materials and Methods

### 2.1. Cows, Housing and Sampling Schedule

In total, 42 Holstein cows in the perinatal period, all multiparous, were used in this study. Samplings for the study began on March 2024 and continued until May 2025, starting from the moment the cows were moved into the maternity pen until 48 h after calving. Samplings were performed twice a day, every morning and evening, and were composed of physical examination, thermography shooting and blood collection, in this respective order.

The cows were housed in a free-barn confinement at the University farm of Obihiro University of Agriculture and Veterinary Medicine, Hokkaido, Japan. TMR was mixed and distributed by a robot. The average number of calvings of the sampled cows was 3.07, and the average milk yield of the farm during the sampling period was 33.2 L/head/day. The ventilation system was turned on 24 h a day.

Farm staff measured the rectal temperature of the late dry period cows once a day in the evening, in order to detect a significant drop in body temperature, indicating that calving was approaching [17,18]. When the temperature drop was observed, the cow was moved to a maternity pen equipped with cameras, where they stayed for at least 24 h after calving. Afterwards, in order to better observe the cows in the peripartum period, they were moved to a hospital pen with other cows that were being treated for diseases. After at least 5 days, the cows were moved to the milking pen, together with all the other lactating cows. The late dry period pen had windows and a gate that granted the animals free access to the outside of the barn, while the maternity and hospital pens did not have access to the outside nor windows close by, but it was close to a gate used by the staff, leading outside.

### 2.2. Physical Examination, Environmental Data and Air Temperature Groups

After restraining the animal, a physical examination was performed by a veterinarian. Data on rectal temperature, heart rate, respiratory rate, rumen motility, ear temperature to the touch and presence of ping sound were examined in order to exclude cows with concomitant diseases.

While performing the physical examination, air temperature and relative humidity were measured using a digital thermometer and hygrometer (CHE-TP1, Sanwa Supply, Okayama, Japan). The data shown by the device after 5 min of turning it on was annotated and used to divide the cows into the two air temperature groups: LT (−1.6 to 14.6 °C) and HT (15.3 to 31.2 °C). Those air temperature intervals were defined with the objective of creating groups as equally sized as possible, without splitting the data from 1 DIM and 2 DIM of the same animal into different groups. By the end, the LT group contained data from 20 cows, while the HT group had 22 cows.

### 2.3. Thermography Photos and Analysis

The thermography shoot was performed immediately after physical examination. Before beginning it, the ventilation was turned off and the windows closest to the cows were closed. The cows were tied to a place away from direct sunlight and natural wind, where they waited standing up for at least 15 min, in order for their skin surface temperature to acclimate to the environment [19,20]. Images of the back side of the ear were taken with a distance between 30 and 50 cm, with the ear as parallel as possible in relation to the camera (Figure 1).

The camera used was a thermographic camera (FLIR ONE Pro-Android, Teledyne Technologies, Wilsonville, OR, USA) mounted on a smartphone (Google Pixel 8, Google LLC, Mountain View, CA, USA), using the first party software designed for that camera (FLIR ONE for Android, Ver. 5.3.6, Teledyne Technologies, Wilsonville, OR, USA). The distance from the camera to the ear was measured using a laser distance measure (Makita LD030P, Makita Corporation, Anjo, Japan).

The images were then analyzed using the software FLIR Thermal Studio Pro (Ver. 2.0.32, Teledyne Technologies, Wilsonville, OR, USA). With the software, an area was manually drawn following the borders of the ear and closed at the thinnest part of its base (Figure 2). Ear tags or holes were excluded from the area when drawing. The temperature data of every pixel was then converted into a Comma Separated Values file, which was used to create a histogram of the temperatures of the ear (Figure 3).

Since the curvature of the inside of the ear is concave and the outside is convex, together with the fact that the inside of the ear has longer strands of hair when compared to the outside, the temperature dissipation levels could be different between both sides [21,22]. Therefore, the present study selected only the back side of the ear for the images.

The temperature profile of each ear did not show a normal distribution. Therefore, the median surface temperature of the ear (MedianEST) was selected for statistical analyses. Using this data, the difference between the MedianEST and the air temperature at the time of sampling (Ear/AirDiff) was also calculated through the formula $\left[MedianEST-air\;temperature=Ear/AirDiff\right].$

### 2.4. Blood Sampling

After the thermography shooting ended, blood was collected by a veterinarian through the tail vein, using vacuum tubes containing blood clot accelerant and serum separating gel (SIM-K0706S-Blue, Sekisui Chemical Co., Ltd., Tokyo, Japan) and was sent to a commercial laboratory (Obihiro Center of Clinical Analysis, Obihiro, Japan) for the estimation of serum iCa2+ concentrations. Cows with serum iCa2+ concentrations below 1.00 mmol/L were considered hypocalcemic [9,10].

### 2.5. Statistical Analysis

Calcium status was categorized as normocalcemia when their serum iCa2+ concentration was equal to or above 1.00 mmol/L, as subclinical hypocalcemia when it was below 1.00 mmol/L without any symptoms at the moment of physical examination, and as clinical hypocalcemia when it was below 1.00 mmol/L concomitant with the presence of symptoms such as paresis, lack of appetite (no interest for newly offered feed) and low ear temperature to the touch [3,4]. In all statistical analyses, the hypocalcemic group included both clinical and subclinical cases.

Each cow had two samples included in their respective air temperature group, one from the first 24 h after calving and another from 24 to 48 h after calving. Since samplings were performed twice a day, samples for each day after calving were picked at random.

The collective data on ear surface temperature did not show a normal distribution after performing a Shapiro–Wilk test (*p* < 0.001), so non-parametric tests were used for the statistical analysis. Spearman’s correlation analysis was performed between air temperature, humidity, rectal temperature, serum iCa2+ concentration, MedianEST and Ear/AirDiff for all cows (Table 1).

U-tests were performed to identify a difference in rectal temperature, humidity and MedianEST between healthy and hypocalcemic cows in each air temperature group separately (Table 2, Figure 4). The MedianEST of healthy cows in both air temperature groups was also compared through a U-test, in order to confirm if the ear surface temperature differs significantly between the two air temperature intervals (Table 3). Finally, logistic regressions were performed to assess the possibility of detecting hypocalcemia in each air temperature group using different combinations of MedianEST, Ear/AirDiff and DIM as covariates (Table 4). Receiver operating characteristic (ROC) analyses were performed to generate an area under the curve (AUC) in order to assess the accuracy of each model.

*p*-values < 0.05 were considered significant, and values between 0.05 and 0.1 were considered to represent a tendency. Statistical analysis was performed using JASP for Windows and for Linux (JASP Team (2024). JASP (Version 0.19.2) [Computer software]).

## 3. Results and Discussion

The number of samples for each group was 40 for LT and 44 for HT, adding to a total of 84 samples. A total of 24 of the cows showed at least one instance of hypocalcemia. Four of the hypocalcemic cows showed symptoms of milk fever, and were treated with intravenous Ca on the day of diagnosis. Only the samples taken before Ca treatment were used in the analysis. The median blood iCa2+ concentration was 1.11 (0.56) mmol/L (median [IQR]) for the normocalcemic cows, and 0.89 (0.4) mmol/L for the hypocalcemic ones.

After acquiring a certain number of samples, it was possible to see that the ear tends to become colder from its extremities towards the base (Figure 3a,b). This may happen because of how the vascularity of the ear is structured, with the auricular vein and the auricular artery branching out into smaller capillary vessels as they approach the borders of the ear [23]. Even in healthy cows, it is presumed that the ear acts as a thermoregulator of the body, so adaptations of the blood flow in the ear are constantly happening [24]. Therefore, instead of focusing on just one area or point, analyzing the ear as a whole could be useful in detecting slight changes near its periphery. Also, well vascularized and warm ears tend to show a histogram closer to a normal distribution (Figure 3c), while cold ears do not (Figure 3d). This difference in pattern may be because of the parts of the ear where blood is flowing in abundance, which would be higher in temperature, and the parts with less blood flow, which would get colder.

When performing Spearman’s correlation analysis (Table 1), air temperature showed a strong positive correlation with MedianEST (*p* < 0.001; r = 0.806) and a weak negative correlation with Ear/AirDiff (*p* < 0.01, r = −0.321). MedianEST also showed a weak positive correlation (*p* < 0.01) with humidity (r = 0.343), rectal temperature (r = 0.322) and serum iCa2+ (r = 0.310). Ear/AirDiff showed a weak positive correlation with serum iCa2+ (*p* < 0.05, r = 0.246). U-tests were performed to observe if the MedianEST of healthy cows are different from the MedianEST of hypocalcemic cows in both air temperature groups (Table 2). The MedianEST of healthy and hypocalcemic cows were not significantly different in HT (*p* = 0.119), but were significantly different in LT (*p* = 0.014). There was also a significant difference in MedianEST between normocalcemic cows in the two air temperature groups (*p* < 0.001; Table 3, Figure 4), highlighting the impact of air temperature when analyzing ear surface temperature. U-tests were also performed using the rectal temperature and humidity at the time of sampling to assess if the effect of those variables in MedianEST differed among normocalcemic and hypocalcemic cows in both air temperature groups, but no significant differences were found (Table 2).

Air temperature has been previously reported to strongly influence ear skin surface temperature [24] and act as a confounding factor when trying to detect hypocalcemia through infrared thermometry [6], but as far as we know, it has not yet been taken into account when developing a model for detecting calcium status. In this study, air temperature showed a strong positive correlation with MedianEST, but when a logistic regression included it as a covariate together with MedianEST for detecting Ca status, the *p*-value for air temperature was much higher than 0.05. The same occurred for humidity when attempting to include it in the model. This likely occurred due to the collinearity between MedianEST, air temperature and humidity (Table 1), which makes it impossible to use them together as independent variables in a logistic regression model. Thus, in order to include air temperature into the model, the two air temperature groups were created and separate logistic regressions were performed for each group (Table 4).

At first, in order to include both air temperature and humidity in the analysis, the groups were to be divided according to the temperature–humidity index (THI) at the time of sampling. However, the THI obtained from three different formulas [25,26,27] showed a strong correlation with air temperature (r ≥ 0.965), and a weak correlation with humidity (r ≤ 331). Therefore, in order to simplify the separation method of the cows and the objectivity of the analysis criteria, only air temperature was used to create the two groups.

The logistic regressions (Table 4) using only MedianEST as a covariate for detecting hypocalcemia generated valid models for both air temperature groups (*p* < 0.05), but the variable performed better at the LT group, which showed a lower Akaike information criterion (AIC) and higher AUC (48.743 and 0.729, respectively). Different combinations of MedianEST, Ear/AirDiff and DIM were also tested, but none of them generated significant models for diagnosing hypocalcemia in the HT group. As for the LT group, aside from DIM by itself (*p* = 0.336), every combination of variables generated a valid model. Ear/AirDiff by itself (*p* < 0.001) had an AIC of 45.341 and an AUC of 0.788. The combination of MedianEST with DIM (*p* = 0.002) resulted in an AIC of 42.543 and an AUC of 0.790, and Ear/AirDiff with DIM (*p* < 0.001) generated an AIC of 43.785 and an AUC of 0.839. Both combinations had the same specificity (0.739), but Ear/AirDiff with DIM had a higher sensitivity (0.765), making it the best performing model overall.

Ear/AirDiff was tested as a variable for detecting hypocalcemia with the hypothesis that hypocalcemic cows would have impaired vasomotor capabilities [28], and with the diminished blood flow, the MedianEST would become closer to the environmental temperature. However, while the logistic regression results were slightly better than MedianEST for LT, with a higher AUC and smaller AIC, it did not generate a valid model for HT. This is likely due to the higher change that occurs in the LT group, since the HT group shows temperatures that are closer to the blood temperature of the cows.

DIM was initially included as a way to mitigate the random effects of the repeated measurements of the cows, but when combined with either MedianEST or Ear/AirDiff, it resulted in better performing models in the LT group, with higher AUC and smaller AIC in both cases. This could indicate that thermoregulation of the ear may vary in how it responds to hypocalcemia during the first 2 days after calving, possibly because of the stress from parturition, and including it as a factor in the model may have corrected the ear surface temperature cut-off points according to the day.

Since parity has been correlated with serum Ca concentrations in the peripartum period [29], it was also tested as a covariate for detecting hypocalcemia, but because its *p*-value was above 0.05 in every combination it was included, it did not result in a useful model. However, when used with MedianEST in the LT group, it showed a tendency (*p* = 0.059) to be a useful variable. Therefore, increasing the number of samples with diverse numbers of calvings, and including parity in the model could improve its general performance in detecting hypocalcemia.

Despite showing promise as being useful for the detection of hypocalcemia in dairy Holstein cows, the present analysis method still lacks real applicability in a commercial farm setting. Keeping the ambient temperature in the range of the LT group, the most accurate range found in this study, could only be realistically achieved in smaller, confined spaces, like a milking parlor. The use of the current method on larger areas would be confined to regions that reach such low temperatures naturally, and only during the cold season. Also, the current ear drawing method still involves a certain degree of subjectivity, such as deciding where the base of the ear is the thinnest, or where the borders of the ears are, due to hair length and heat radiation (Figure 2). Different shapes and attachment locations of ear tags could also affect the analyzable area of the ear, and despite closing down the shutters of the confinement area and shutting down the ventilation system, wind could still be entering the area and affecting the surface temperature of the ears [30]. Furthermore, this method of analysis requires manual drawing and cannot be performed at cow-side, only on a personal computer equipped with the necessary software.

While dividing the cows into air temperature groups helped to show how much it influences the surface temperature of the ear, as well as how hotter environments can diminish the efficacy of using ear surface temperature as a predictive variable for hypocalcemia, further study is necessary in order to determine the best air temperature thresholds for accurate diagnosis. Therefore, even though this analysis method requires simplification, or even automation, as well as a realistic way of subjecting the cows to a colder environment, ear surface temperature detected through infrared thermography shows promise as a support variable for detecting both clinical and subclinical hypocalcemia in Holstein dairy cows.

## 4. Conclusions

This study was conducted with the objective of assessing the capability of infrared thermography in identifying changes in the surface temperature of the ear depending on the iCa2+ status of Holstein dairy cows, as well as if said data can help in the detection of hypocalcemia. The results suggest that the ear surface temperature lack applicability in warmer environments, but can be useful in colder ones. However, practical application and replicability still have room for improvement. In the future, automation of this analysis method could be developed if technologies such as machine learning were applied for image recognition of the ear and automatic drawing of the analysis area.

## Figures and Tables

**Figure 1 animals-15-02055-f001:**
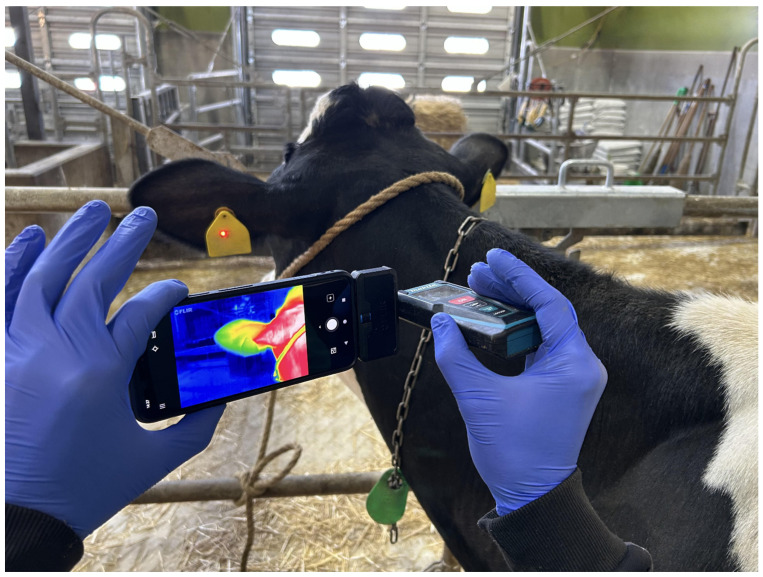
Shooting method for the thermography of the ears. After confirming the distance with a laser distance measure (30–50 cm), the image was taken using the first party application FLIR ONE (Ver. 5.3.6), with the thermographic camera mounted on a smartphone positioned as parallelly as possible to the back of the ear.

**Figure 2 animals-15-02055-f002:**
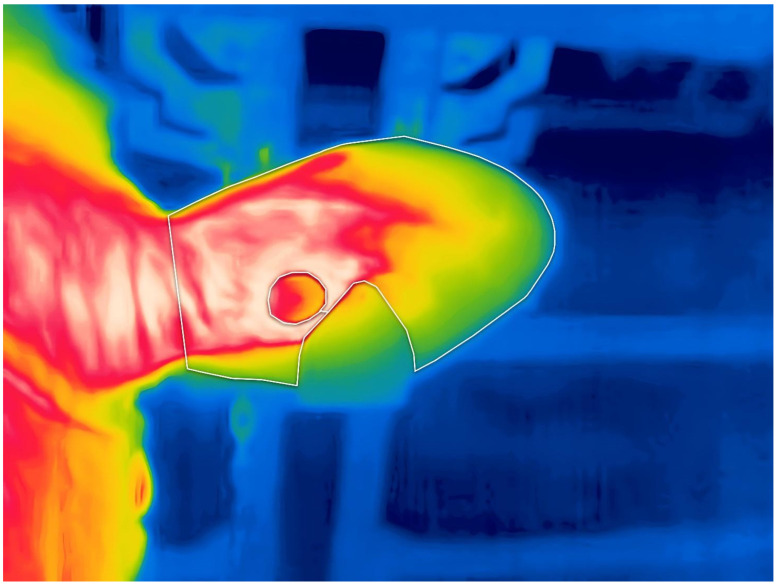
Drawing method used on the thermographic images to extract the temperature profile of the ear. Using the software FLIR Thermal Studio Pro (Ver. 2.0.32), the polygon tool was selected and an area was drawn around the edges of the ear, finishing at the point where the base of the ear was the thinnest. Ear tags were excluded from the drawn area.

**Figure 3 animals-15-02055-f003:**
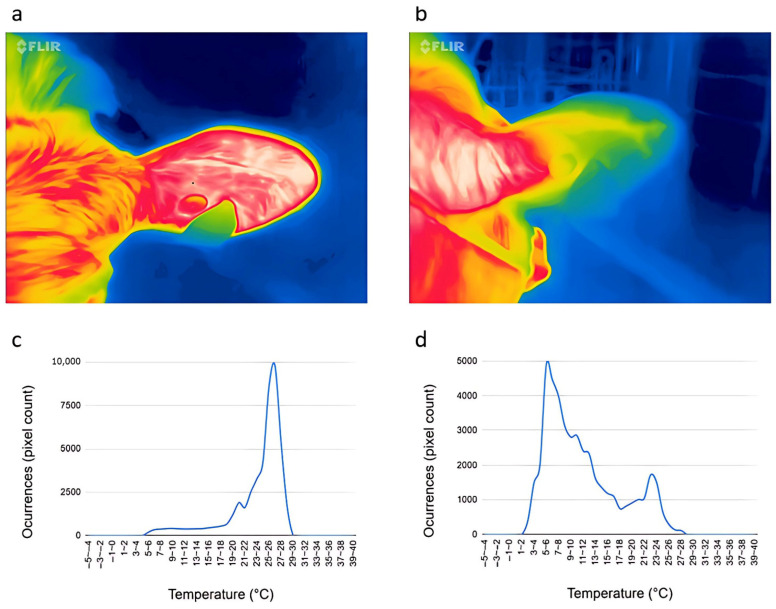
Thermographic profiles of the same cow at different sampling moments. One ear was warm to the touch at the moment of physical examination (**a**), while the other was cold (**b**). Below are histograms representing the temperature profiles of the warm ear (**c**) and the cold ear (**d**).

**Figure 4 animals-15-02055-f004:**
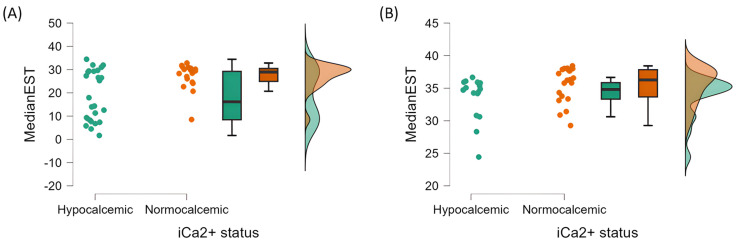
U-tests between the median ear surface temperature (MedianEST) of normocalcemic and hypocalcemic cows in two different air temperature groups: lower air temperature group (LT, panel (**A**)) and higher air temperature group (HT, panel (**B**)).

**Table 1 animals-15-02055-t001:** Spearman’s correlations between air temperature, humidity, rectal temperature, iCa2+, MedianEST and Ear/AirDiff.

Variable ^1^	Air Temperature	Humidity	Rectal Temperature	iCa2+	MedianEST	Ear/AirDiff
1. Air temperature	__					
2. Humidity	0.019	__				
3. Rectal temperature	0.354 ***	0.252 *	__			
4. iCa2+	0.178	−0.075	0.071	__		
5. MedianEST	0.806 ***	0.343 **	0.322 **	0.310 **	__	
6. Ear/AirDiff	−0.321 **	0.048	−0.148	0.246 *	0.198	__

^1^ MedianEST = median ear surface temperature (°C); Ear/AirDiff = difference between the median ear surface temperature and the air temperature (°C). * *p* < 0.05; ** *p* < 0.01; *** *p* < 0.001; *n* = 84.

**Table 2 animals-15-02055-t002:** U-tests between the MedianEST of normocalcemic and hypocalcemic cows in two different air temperature groups.

Groups and Variables ^1^	U	df	*p*-Value
Rectal temperature-LT	233.500		0.301
Humidity-LT	199.000		0.935
MedianEST-LT	106.000		0.014
Rectal temperature-HT	183.500		0.203
Humidity-HT	256.000		0.669
MedianEST-HT	171.000		0.119

^1^ MedianEST = median ear surface temperature (°C); LT = lower air temperature group (air temperature −1.6 to 14.6 °C); HT = higher air temperature group (air temperature 15.3 to 31.2 °C).

**Table 3 animals-15-02055-t003:** U-test between the MedianEST of normocalcemic cows in two different air temperature groups.

Goups and Variables ^1^	U	df	*p*-Value
LT and HT-MedianEST	409.000		<0.001

^1^ MedianEST = median ear surface temperature (°C); LT = lower air temperature group (air temperature −1.6 to 14.6 °C); HT = higher air temperature group (air temperature 15.3 to 31.2 °C).

**Table 4 animals-15-02055-t004:** Logistic regressions for two air temperature groups using iCa2+ status (serum iCa2+ concentration above or below 1 mmol/L) as the dependent variable and various combinations of ear surface temperature, difference between ear and air temperatures and days in milk as covariates.

Models Described by Groups and Variables ^1^	Sensitivity	Specificity	^2^ AUC	^3^ AIC	Odds Ratio	95% Confidence Interval (Odds Ratio Scale)	*p*-Value
LT							
MedianEST	0.706	0.609	0.729	48.743	1.129	1.032–1.235	0.002
Ear/AirDiff	0.765	0.609	0.788	45.341	1.160	1.048–1.284	<0.001
DIM	0.000	1.000	0.577	57.624	0.538	0.151–1.917	0.336
MedianEST + DIM	0.647	0.739	0.790	42.543	__	__	0.002
Ear/AirDiff + DIM	0.765	0.739	0.839	43.785	__	__	<0.001
HT							
MedianEST	0.840	0.316	0.640	59.435	1.209	0.994–1.470	0.029
Ear/AirDiff	1.000	0.000	0.509	64.143	1.015	0.865–1.191	0.856
DIM	1.000	0.000	0.523	64.084	0.831	0.252–2.743	0.761
MedianEST + DIM	0.760	0.316	0.657	61.250	__	__	0.085
Ear/AirDiff + DIM	1.000	0.000	0.528	66.051	__	__	0.939

^1^ LT = lower air temperature group (air temperature −1.6 to 14.6 °C); HT = higher air temperature group (air temperature 15.3 to 31.2 °C); MedianEST = median ear surface temperature (°C); Ear/AirDiff = difference between the median ear surface temperature and the air temperature (°C); DIM = days in milk, being either the first day (0 to 24 h) or the second day (24 to 48 h) after calving; ^2^ AUC = area under the curve after ROC analysis; ^3^ AIC = Akaike information criterion.

## Data Availability

All relevant data are within the manuscript and its Appendix A.

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
