# Peer review of "Evaluation of Ear Thermographic Imaging as a Potential Variable for Detecting Hypocalcemia in Postpartum Holstein Dairy Cows"

_animals, 2025, doi:10.3390/ani15142055_

Round 1
Reviewer 1 Report
Comments and Suggestions for Authors
in attachment

Reviewer 2 Report
Comments and Suggestions for Authors
On dairy farms, monitoring postpartum calcium levels is essential for preventing or diagnosing clinical or subclinical hypocalcemia and associated conditions. It is therefore essential to develop methods that allow dairy farm staff without training in veterinary medicine to do this autonomously, e.g., by using new technologies, whether or not associated with AI.
Thermography, also because the equipment is becoming less expensive, is beginning to be used more for various purposes in farm health monitoring, and here choosing the ear makes a lot of sense, taking into account the symptomatological picture of hypocalcemia.
In this study, unfortunately, the interference of environmental temperature, especially on hotter days, complicated the effectiveness of this method, but perhaps this can be overcome in further research.
If I may, I leave a suggestion to the authors: if you haven't done so yet, complement your data with pupillary dilation (and also reflex), as many dairy cattle practitioners consider pupil dilation to be a good indicator of hypocalcemia. A flashlight is enough, but you can also take a photo for later analysis by software.
Reviewer 3 Report
Comments and Suggestions for Authors
In Materials and Methods:
I´m missing an exact time of the first blood collection. Was it done as a part of physical examination after the birth?
In Results and Discussion:
I´m missing mean Ca levels in Ca groups. Could they be affected by Ca of cows with clinical hypocalcemia?
Round 2
Reviewer 1 Report
Comments and Suggestions for Authors
Now, manuscript is OK